# Antibiotic Therapy for Prosthetic Joint Infections: An Overview

**DOI:** 10.3390/antibiotics11040486

**Published:** 2022-04-05

**Authors:** Benjamin Le Vavasseur, Valérie Zeller

**Affiliations:** 1Referral Center for Complex Bone and Joint Infections, Diaconesses Croix Saint-Simon Hospital, 75020 Paris, France; levavasseur@hopital-dcss.org; 2Department of Internal Medicine and Infectious Diseases, Diaconesses Croix Saint-Simon Hospital, 75020 Paris, France

**Keywords:** prosthetic joint infections, antibiotic therapy, monotherapy and combination therapy, pharmacokinetic/pharmacodynamics optimization, therapeutic drug-monitoring, patient education, adverse events

## Abstract

Prosthetic joint infection (PJI) is a severe complication after arthroplasty. Its management combines surgical intervention, whose type depends on the clinical situation, and prolonged high-dose antibiotics adapted to the responsible microorganism(s) and the patient. Antibiotics are only one part of the therapeutic regimen and are closely related to the surgical strategy. Their efficacy depends to a large extent on the choice and quality of the surgical procedure, and the quality of the microbiological diagnosis. Although guidelines have been published, many aspects of antibiotic therapy remain poorly established. Choosing the optimal agent(s) is one aspect, with others being optimization of drugs’ pharmacokinetic/pharmacodynamic parameters, the choice of administration route, use of monotherapy or combination regimens, therapeutic drug-monitoring and patient education to improve compliance and tolerance. Herein, we address PJI management based on recent literature data, guidelines and the experience of our referral center for complex bone-and-joint infections.

## 1. Introduction

Joint arthroplasty is now a very common and successful surgical procedure to restore joint function. Nearly 200,000 total hip replacements are performed annually in France. It is estimated that, by 2050, this number will increase 42–112.3% [1]. This operation also carries risks, however, among which prosthetic joint infection (PJI) is the most severe, carrying a high economic and human burden. The PJI rate is 0.5–2% for primary knee, hip and shoulder replacements [2,3]. That rate is much higher for revision arthroplasty, reaching up to 15% [4].

PJI treatment is complex because it deals with a large number of different clinical contexts, relying on specific surgical and medical strategies, which have to be adapted to the patient. A multidisciplinary approach, with strong collaboration among all medical and surgical personnel involved, is essential. Antibiotics represent only a part of PJI management and are closely linked to the surgical strategy applied. Their efficacy is highly dependent on the choice and quality of the surgical procedure, and on the quality of the microbiological diagnosis. International guidelines were published nearly 10 years ago [5]. As stressed in the guidelines introduction, many of those recommendations were based on expert opinions. Indeed, the number of high-evidence studies to support a specific recommendation was limited and diverse practice patterns for a given clinical setting seemed to obtain similar outcomes. This review is not intended to be a substitute for those guidelines. Herein, we first develop more practical aspects and key points of antibiotic therapy for PJIs, and then detail the specific management of various microorganisms. It brings together recent findings on the subject and the extensive experience of our reference center for complex bone and joint infections.

Clinical diagnosis of PJIs can be challenging because of the indolent course of chronic infections, difficulties culturing low-virulence bacteria, and the wide variety of microorganisms and antibiotic-resistant phenotypes that can be involved [6,7]. Precise identification of the microorganism(s) and their antibiotic susceptibilities are crucial to successfully treating these patients [8].

Optimal curative treatment of PJI combines surgery and prolonged, high-dose, antibiotic therapy. Two very different surgical strategies are applied, depending on the clinical picture: prosthesis retention with débridement, synovectomy and exchange of the mobile elements for acute infection, while biofilm formation is still ongoing; or prosthesis removal followed by a new arthroplasty during one- or two-stage exchange for chronic infections.

## 2. Initial IV Antibiotics and Extended Infusion

PJI treatment requires optimal antibiotic coverage, especially during the early phase, i.e., the first postoperative weeks. Although the bacterial load is reduced by surgery, high-dose antibiotics and optimization of PK/PD parameters achieve better infection control [9] and lower the risk of bacterial resistance emergence. Furthermore, to avoid selection of resistant strains and subsequent treatment failure, initial treatment should include bacterial load-independent drugs with low risk of resistance development, such as β-lactams, vancomycin or daptomycin. Intravenous administration allows use of higher doses, bypasses the intestinal and hepatic first-pass effect that hinders bioavailability of certain drugs [10], and avoids malabsorption related to gastrointestinal intolerance and concomitant medications (heavy metals, such as iron, which reduce drug absorption) that are frequently prescribed to PJI patients, around 70 years old [11]. All these advantages are strong arguments supporting initial IV antibiotics.

To optimize the efficacy of time-dependent antibiotics, such as β-lactams, vancomycin and clindamycin, whose activities are linked to the time during which their concentrations exceed the pathogen’s minimal inhibitory concentration(s) (MIC(s)), continuous or extended IV infusion is appropriate.

The results of a recent, prospective, comparative study on continuous or 3 h vs. intermittent cloxacillin infusion showed >90% probability of target attainment, i.e., 100% of the time at a MIC < 0.5 mg/L, for the continuous-infusion group [12].

We reported the feasibility, efficacy and safety of continuous cefazolin, clindamycin, vancomycin or ceftazidime [13,14,15,16] infusions to treat bone-and-joint infections (BJIs) with very good tolerance and high cure rates, including for patients with difficult-to-treat infections.

In a recent, prospective PK study on the influence of the administration route on the magnitude of clindamycin–rifampicin interaction, we also showed that continuous clindamycin infusion reached therapeutic plasma concentrations (3–8 mg/L), even combined with rifampicin, a strong inducer of hepatic enzymes, thereby significantly decreasing plasma clindamycin concentrations [10]. When combined with rifampicin, intermittent clindamycin infusion trough concentrations (C_min_ 0.37 mg/L) were very low, dramatically below the target concentration [17].

The use of continuous meropenem administration was evaluated in some trials, whose results indicated improved PK efficacy [18,19], better bacteriological eradication [19] and higher clinical cure rates [20].

Continuous IV infusion requires an initial antibiotic-loading dose, started at the same time as the continuous infusion, to rapidly attain the target plasma concentration. Moreover, adequate preparation (dilution, choice of the solution, etc.) and antibiotic administration (choice of the infusion device, solution volume, time of infusion) by experienced teams, during hospitalization and outpatient therapy, are essential to assure drug stability and antibacterial activity [21].

Little is known about the bone antibiotic concentration obtained after continuous antibiotic infusion. In our cohort study on 100 patients treated with continuous IV cefazolin infusion, the determinations of eight patients’ bone cefazolin concentrations revealed broad variability and a median bone/plasma ratio of 25% [13].

## 3. Oral vs. IV Therapy

How long the initial IV regimen should be administered remains controversial. The trend is to shorten IV therapy to avoid catheter-related complications, shorten the hospital stay, lower related costs and improve patient comfort [22].

The British randomized-controlled Oral Versus IntraVenous-Antibiotic (OVIVA) trial compared patients treated for nonsevere BJIs with 6 vs. 1 week of IV antibiotics, then continued orally. The study population consisted of a variety of BJIs (osteomyelitis, arthritis, PJI, etc.), with and without foreign material, which were treated with either implant retention or removal. After 1 year of follow-up, no difference between the two groups was observed in terms of treatment failures, 14.6% vs. 13.2%, respectively [23]. Authors of a Cochrane meta-analysis concluded similarly [24]. Authors of several studies reported good outcomes after debridement and implant retention (DAIR) for acute PJI with oral antibiotic regimens, including rifampin and levofloxacin or ciprofloxacin, for staphylococcal [25], and fluoroquinolones for Gram-negative infections [26]. Those results provide arguments supporting the use of certain oral antibiotics for selected patients (Table 1).

Several factors have to be taken carefully into consideration to determine when treatment can be continued orally (Figure 1). First, bacteria must be susceptible to oral antibiotics that are well tolerated and easily absorbed by the intestine. Furthermore, drugs must not or only poorly be metabolized in the intestine and liver, to ensure therapeutic plasma and tissue concentrations. Pathogen-specific regimens are detailed below. In our center, IV treatment is maintained for at least 1 week after surgery for susceptible microorganisms. If the local and general infection evolution is favorable, antibiotic gastrointestinal tolerance is good and treatment modalities are understood by a reliable patient, an oral regimen, individualized considering the above-listed precisions, can be started. A follow-up consultation to assess treatment compliance, effectiveness and tolerance should be scheduled 1 or 2 weeks after the patient’s discharge [27]. The scheduling of subsequent visits will depend on treatment duration and tolerance. Regular biological monitoring is also indicated. Seaton et al. emphasized that IV and prolonged-and-complex oral antibiotic administration and follow-up can be managed within an outpatient antibiotic therapy (OPAT) service, which they named complex outpatient antibiotic therapy (COpAT) [28].

For patients with difficult-to-treat or resistant bacteria, complex surgeries including extensive bone reconstruction and/or gastrointestinal intolerance or malabsorption, IV antibiotic administration will be prolonged or even maintained for the entire treatment duration. After hospital discharge, treatment can be administered IV by either OPAT services or experienced rehabilitation teams.

## 4. Patient Education

Poor treatment adherence and self-medication are two of the most important problems of antibiotic misuse. Adherence has been specifically studied in chronic but not acute diseases, such as infectious diseases. A meta-analysis found that 38% of patients forget to take at least 1 antibiotic dose [29]. Recent findings confirmed that patient education significantly enhanced adherence to prescribed antibiotics and reduced wastage [30]. During hospitalization in our center, patients receive individualized counseling, and are given written information about antibiotic-treatment modalities, dose, duration and possible adverse events (AEs).

## 5. Antibiotic Treatment Duration

Prolonged antibiotic use is required. Administration duration remains controversial and ranges between 6 and 12 weeks, according to the medical–surgical strategy (implant retention vs. removal) and teams. Duration is longer for DAIR strategies and reportedly can be <2 weeks after two-stage arthroplasty [31].

Very few randomized trials have explored that situation. Most included small numbers of patients. Different surgical strategies were applied. According to a Spanish study that included 63 patients with acute staphylococcal PJIs managed with DAIR and combination levofloxacin–rifampicin found that 8 weeks of antibiotics was not inferior to 3–6 months [32]. A Swiss randomized trial, comparing 4 vs. 6 weeks of antibiotics to treat various BJIs, included 38 PJIs managed with implant removal by two-stage exchange; no between-group difference was observed with respective microbiological cure rates of 98% vs. 94% after median follow-up of 2.2 years [33]. Very recently, a large French open-label, randomized, noninferiority trial compared 6 vs. 12 weeks of antibiotics for patients with PJIs (205/group) [34]. Notably, noninferiority of 6-week therapy was not confirmed, with 18% of the 6-week group experiencing persistent infection vs. 9% of the 12-week group. Subgroup analyses revealed important differences according to the surgical strategy: 6-week therapy resulted in significantly more unfavorable outcomes than 12-week therapy for DAIR-managed patients (30.7% vs. 14.5%, respectively); outcomes did not differ for patients who had undergone one-stage exchange (4% vs. 2.8%), while those who underwent two-stage exchange and received 6 weeks of treatment also had more events (15% vs. 4.9%).

Taken together, many questions concerning antibiotic duration for PJIs persist; it has to be adapted to the surgical strategy. Recent data underline the importance of longer treatment for prosthesis retention. We recommend treating acute DAIR-managed PJIs for 12 weeks, if the regimen is well tolerated. Chronic PJIs with prosthesis removal managed with one- or two-stage exchange arthroplasty, exceptionally with resection arthroplasty, can be treated for 6 weeks postoperatively. For cases at high risk of relapse, e.g., patients receiving chemotherapy or taking immunosuppressants, with CHILD B or C cirrhosis, PJI involving irradiated bone or extensive osteomyelitis or requiring large bone grafts, or patients with relapsing PJI previously having been treated with exchange arthroplasty, more prolonged treatment lasting 12 weeks should be considered (Figure 2). The type of microorganism might also influence treatment duration. Clinical experience suggests, for example, that *S. aureus* PJIs require longer treatment than very susceptible streptococcal or enterobacterial infections. Future randomized trials on these different and specific situations are needed to further clarify treatment duration.

## 6. Antibiofilm-Active Drugs

PJI pathophysiology is characterized by biofilm formation, a universal and major bacterial survival mode in which bacteria form a microbe-derived sessile community attached to a surface. Biofilm is composed of an 80% polysaccharide layer (glycocalyx) in which the bacteria are embedded. They adhere irreversibly to the prosthesis, slow their metabolism, multiply little and their growth rate and gene-transcription phenotypes are modified. Biofilm formation protects them from the host’s cellular response thereby enabling antibiotic resistance and tolerance [35]. Treatment of biofilm-associated infections is difficult, especially when the infected implant is retained. Biofilm logarithmically decreases antibiotic efficacy [36]. The most efficient strategy is to remove the foreign material. Biofilm eradication by antibiotic therapy alone has proven to be difficult [37,38].

Rifampicin antibacterial activity against staphylococcal infections in animal models is concentration-dependent, with the area under the time–concentration curve (AUC)/MIC and peak concentration/MIC ratios being predictive of its activity [39]. The drug remains active against slow-growing bacteria in biofilm [40]. Results of in vitro and in vivo experiments showed good activity against staphylococcal biofilm infections of rifampicin combined with many other antibiotics (linezolid, cefazolin, oxacillin, vancomycin, gentamicin, azithromycin, ciprofloxacin or fusidic acid) [41]. Resistance can emerge rapidly on monotherapy [42], however, especially when the bacterial load is high. Therefore, rifampicin should always be administered with another antistaphylococcal drug and started a few days after surgery, once the bacterial burden has been reduced [40].

The optimal rifampin dose remains a matter of debate. Recommendations and practices vary according to country and team. Authors of a recent, retrospective, single-center study on 411 patients with acute or chronic staphylococcal PJIs treated with various strategies (prosthesis retention, exchange arthroplasty, others) found that rifampicin doses <10 mg/kg/day were not less effective than higher doses (10–20 mg/kg/day and >20 mg/kg/day) [43]. However, only 27 patients received low doses and the rifampicin unit dose was not specified.

Rifampicin has complex PK properties. Its intestinal absorption varies with food intake. It is metabolized in the intestinal tract and the liver. It is a major cytochrome P450 and intestinal transporter inducer, responsible for important drug interactions, and even induces its own transformation and elimination. Marked interindividual AUC and plasma-concentration variations have been observed [44,45]. The abovementioned absorption and metabolic processes, but also genetic polymorphism of drug influx and efflux transporter genes, influence rifampicin bioavailability [45].

Taking into account all these drug characteristics and fragility, we recommend avoiding low doses, giving a 600 mg unit dose, twice a day, except for very underweight (<50 kg) or overweight (>90 kg) patients, for whom the dose must be adapted. Studies and recommendations on tuberculous treatment underscored that the higher rifampicin dose obtained better efficacy with no additional toxicity [46]. If available, monitoring of the peak rifampicin concentration can help adjust the dose. Because of the drug’s autoinduction, such monitoring should be done at least after 1 week of treatment. For tuberculosis, the target peak concentration is ≥8 mg/L [47]. The clinical pertinence of that monitoring has to be evaluated for the treatment of staphylococcal PJI.

Gastrointestinal AEs are frequent (20–30%), especially during the first week of treatment. Initial IV administration can help avoid drug underexposure during the early postoperative phase. In their prospective observational study on 46 patients treated for BJIs, Roblot et al. [44] found no significant association between gastrointestinal AEs and hepatotoxicity, and rifampicin trough-and-peak concentrations. Median trough concentration was lower but not significantly different for the group without AEs (0.6 vs. 1 mg/L; =0.94). Early outpatient consultation after discharge to assess tolerance, compliance and drug monitoring can help improve treatment adherence.

Other drugs, such as linezolid, daptomycin and vancomycin, also have antibiofilm antibacterial activity in vitro [41]. To date, large clinical studies are lacking and their precise roles remain to be defined.

## 7. Bone Penetration

Antibiotic bone diffusion is another important parameter of PJI treatment, especially for chronic infections with bone involvement. The bone/plasma ratio has been reported in several publications [48,49,50,51]. Most bone/plasma antibiotic ratios range between 0.1 and 0.3 [50,51]. A few, such as fluoroquinolones, rifampicin and macrolides, have better ratios. However, many studies on bone antibiotic concentrations suffered from major methodological limitations. First, the sample size was often small. Second, bone antibiotic concentrations were usually determined in healthy patients undergoing joint arthroplasty who had received a single antibiotic administration with various doses. Moreover, extracellular/plasma- and intracellular/plasma-concentration ratios were often not distinguished [52]. Many studies are old and had applied a variety of methods to measure bone concentrations. Results might differ if those studies were to be repeated now [50,53]. Furthermore, bone/plasma-ratio data are often not informative concerning antibiotic activity [52].

Finally, bone penetration is one among other factors to consider when choosing antibiotics to treat PJIs. MIC of the involved microorganism, the drug’s PK/PD properties, activity on biofilm-embedded microorganisms, drug toxicity and clinical study results are other major parameters that guide treatment choice.

## 8. Therapeutic Drug Monitoring (TDM)

These difficult-to-treat PJIs require prolonged high-dose antibiotic administration. They often develop in old and/or obese patients with multiple comorbidities and polypharmacy use, which can modify drug PKs. During the postoperative phase, several metabolic disturbances can occur (variations of the volume of distribution, kidney function, drug interactions, etc.). Drug AEs are also frequent. Multiresistant microorganisms can be involved, thereby limiting available therapeutic options. For all these reasons, antibiotic optimization and adjustment to each patient’s characteristics, underlying conditions and drug tolerance, and microorganism susceptibility, are fundamental. Determination of plasma drug concentrations, more commonly known as TDM, is one of the ways to optimize antibiotic use, along with MIC determination and high-dose prolonged or continuous infusion with time-dependent antibiotics.

Vancomycin has a low therapeutic index. In our experience, continuous, IV, high-dose vancomycin combination therapy is effective against methicillin-resistant staphylococcal PJIs. TDM is necessary throughout treatment to limit nephrotoxicity observed in one-third of the patients [15].

Beta-lactams have a large therapeutic index, usually not indicating TDM. However, in complex situations involving multiresistant or difficult-to-treat microorganisms, in multioperated and/or relapsing-PJI patients or those with renal insufficiency or obesity, TDM helps optimize drug administration by limiting over- and underdosing, and reach target concentrations. In a prospective comparative study on 52 patients, Gomez-Junyent et al. compared the outcomes of fluoroquinolone-resistant and susceptible *Pseudomonas aeruginosa* BJIs [54]. Prolonged continuous β-lactam infusion TDM-adjusted, with/without colistin, was used to treat the resistant strains. Failure rates for resistant and susceptible strains were comparable (21% vs. 16.6%, respectively). A French BJI team reported individualized PK/PD parameters of suppressive, β-lactam doses administered subcutaneously to 10 patients with PJIs [55]. The dose interval could be prolonged from twice daily to thrice weekly for some patients; infection was controlled in all but one. Those experiences and ours [13,16,56] provide arguments supporting TDM to manage complex PJIs.

Antibiotic concentration measurements can also reveal or confirm drug interactions. Based on a large retrospective cohort study on 70 BJI patients treated with continuous IV clindamycin combined with rifampicin or without, we previously described a 40% rifampicin-induced decrease in the median plasma clindamycin concentration [14]. More recently we published our results of a prospective, comparative study undertaken to characterize the influence of the clindamycin-administration route, i.e., oral vs. IV, on its PK interaction with rifampicin [11]. Combined with rifampicin, clindamycin clearance was seven-fold higher when taken orally than given IV—significantly higher than initially hypothesized. The very low median clindamycin peak (1.53 mg/L) and trough (0.18 mg/L) concentrations and 10% drug bioavailability in the combination therapy group, question the use of this oral combination therapy for difficult-to-treat infections, such as BJIs. However, continuous, IV clindamycin administration achieved effective plasma concentrations when combined with rifampicin. The observed broad interindividual variability of clindamycin steady-state concentrations warrants TDM to adjust the dose.

No data on TDM of fluoroquinolones in patients with BJIs have been published, but fluoroquinolone TDM is used to optimize treatment of multiresistant tuberculosis [57]. High interindividual variability of plasma concentrations of these drugs was observed that justifies TDM in complex situations.

To conclude, TDM helps optimize antibiotic therapy, especially in complex microbiological or PK settings. However, it is only an evaluation of the bone antibiotic concentration and not an assessment of antibiotic activity; it is, nonetheless, an important parameter for an individual therapeutic approach. Large studies are needed to confirm that TDM improves PJI treatment efficacy and tolerance.

## 9. Empirical Antibiotic Regimens

Empirical antibiotic therapy, usually combining a broad-spectrum β-lactam with good activity against Gram-negative rods, such as piperacillin–tazobactam or –cefepime, and a drug effective against Gram-positive cocci, including methicillin-resistant staphylococci, such as daptomycin or vancomycin, is often—and certainly too often—used at treatment onset, while waiting for the results of intraoperative sample cultures. Different clinical contexts should be distinguished to better discern the indication of such broad-spectrum therapy, which is expensive, markedly impacts human-flora ecology and leads to drug-related toxicity [58]. Thorough analysis of each patient’s history is essential for clinicians managing these infections, to guide the choice of the initial empirical regimen.

For early postoperative PJIs, the wide range of microorganisms that can be involved, a high polymicrobial infection rate and urgent surgery, warrants the use of such a broad-spectrum regimen [8,59].

On the contrary, acute hematogenous PJIs are almost always (99%) monomicrobial. MSSA and streptococcal infections are by far the most frequent pathogens isolated. Cefazolin or cloxacillin is the first-line antibiotic in this context. Third-generation cephalosporins are recommended when Gram-negative rods are suspected. Search for an infection source is an additional step to orient treatment choice [8].

Finally, for late chronic PJIs, urgent surgical intervention is not needed. Management starts with confirming PJI and isolating the involved microorganism(s). In our experience, the choice of the initial antibiotic regimen is guided by those microbiological results and previous intraoperative sample cultures, if the patient has been managed in another center. The initial treatment also depends on the number of previous same-site operations and the surgical strategy applied (exchange arthroplasty, or definitive or temporary implant removal). This type of PJI often harbors low-virulence microbes, such as coagulase-negative staphylococci (CNS), with a high rate of methicillin-resistant *Staphylococcus epidermidis*, but also anaerobic bacteria, for example, *Cutibacterium acnes* [8]. Broad-spectrum combination therapy is not indicated systematically.

For critically ill patients, appropriate broad-spectrum antibiotic therapy is usually initiated rapidly, in combination with an aminoglycoside to enhance the antibacterial activity, unless the patient has developed severe renal insufficiency.

## 10. Pathogen-Specific Antibiotic Therapies

Once microbiological results become available, it is essential to tailor the antibiotic regimen from empirical coverage to pathogen-specific treatment (Table 2).

Clinical practice guidelines from different countries, published during the last 10 years [5,60,61], help clinicians treat these complex infections. Antimicrobial agents for common PJI-causing microorganisms are detailed in those recommendations. Our intention in the following paragraphs is not to recommend treatments but to highlight the points we think essential to achieve the highest treatment success rate. Above, we discussed initial IV-therapy duration, and underscored the contributions of patient education and an early, post discharge follow-up consultation to assess oral treatment compliance, efficacy and tolerance.

### 10.1. Staphylococcal PJIs

Staphylococci are by far the most common microorganisms isolated from PJIs, with *S. aureus* and *S. epidermidis* being the most frequent species [8].

Numerous authors observed and analyzed the outcomes of DAIR-treated MSSA PJIs treated with prosthesis retention. The treatment of choice for methicillin-susceptible staphylococcal PJIs is initial IV antistaphylococcal β-lactam, oxacillin or cefazolin [5,25,60]. Prolonged or continuous infusion of high doses optimizes the drug’s time-dependent activity and is well tolerated [13]. If the local and general evolution is favorable 3–5 days after surgery, rifampicin (IV or oral) should be added. The crucial role of rifampicin in treating MSSA implant-associated infection, its complex PKs and the drug’s inducer properties, and its AEs were addressed above. Notably, a recent randomized controlled trial [62] and a systematic review and meta-analysis [63] tempered rifampicin’s “star” position for PJI treatment. Initial IV β-lactam–rifampicin is maintained around 1 week, and then an oral regimen can be started, if evolution remains favorable, gastrointestinal tolerance is good and treatment modalities are completely understood by a reliable patient. The first-choice oral therapy is levofloxacin (750 mg/day to 500 mg twice a day) and rifampicin [25,64], a combination that causes frequent AEs [64,65].

In a retrospective comparative cohort study, Vollmer et al. analyzed the safety and tolerability of fluoroquinolones in 156 patients with DAIR-treated *S. aureus* PJIs [66]. The unplanned drug discontinuation rate for the patients receiving levofloxacin–rifampicin was significantly higher than that of patients given nonfluoroquinolone-containing regimens (27.5% vs. 4.2%, respectively). The former group also had more severe AEs (7.5% vs. 1.5%). Fluoroquinolones were associated with tendinopathy, myalgias, arthralgias and/or nausea. Rifampicin also required discontinuation (54%), especially during the early IV treatment phase. The authors and the associated editorial commentary [67] insisted on careful patient selection for oral fluoroquinolone therapy, to help them to better tolerate the treatment and accompany them in a COpAT. Alternative nonfluoroquinolone-based regimens have to be considered, and evaluated for efficacy and safety.

When rifampicin use is not possible, alternative treatments are oral regimens combining levofloxacin with clindamycin, minocycline or co–trimoxazole. Monotherapy with those drugs has not yet been validated. However, Infectious Diseases Society of America guidelines recommend cefazolin or oxacillin IV monotherapy for the entire treatment duration [5].

Alternative nonfluoroquinolone regimens are IV rifampicin–clindamycin or –cefazolin with good efficacy and tolerance based on our large, retrospective cohort studies [13,14]. Oral rifampicin–clindamycin achieves very low or undetectable clindamycin trough concentrations with high risks of failure and development of antibacterial resistance [11,17,68]. We recommend not prescribing that combination to treat complex PJIs. However, Bonnaire et al. gave that oral combination with clindamycin to 46 patients for staphylococcal BJIs and erythromycin-resistant but clindamycin-susceptible strains [69]. Twenty patients were managed for PJI, 37 were receiving oral clindamycin–rifampicin. The intention-to-treat cure rate was 64.7%, with 84.6% per-protocol successes. No relapse with a clindamycin-resistant strain was observed. More generally, in their experience, clindamycin-combination therapy appeared to be effective against erythromycin-resistant, lincosamide-susceptible staphylococcal BJIs [69]. According to a 2019 French national audit on the management of PJIs, clindamycin was a frequent choice, when fluoroquinolones or rifampicin were not available [70]. Other oral rifampicin combination partners include co–trimoxazole, minocycline or doxycycline [5] and linezolid. As emphasized by Vollmer et al., comparative analysis of outcomes of DAIR-managed PJIs treated with oral anti-staphylococcal agents is an important area of future exploration to determine whether nonfluoroquinolone-based antibiotics achieve similar efficacy, with fewer AEs [66].

Concerning methicillin-resistant *S. aureus* (MRSA) and CNS, data from randomized trials or large cohort studies are lacking. International guidelines recommend vancomycin as the preferred agent [5]; daptomycin or linezolid are alternatives. MRSA was associated with treatment failure [71]. Methicillin-resistance is definitely higher in *S. epidermidis* strains than *S. aureus* [6,8,72]. Another important factor is the presence of various resistance patterns in a given patient’s staphylococcal PJI population. Antibiotic-susceptibility tests must be performed on multiple isolates to search for multiresistant isolates that could contribute to treatment failure [73].

To optimize vancomycin therapy, continuous IV, high-dose vancomycin-combination therapy is the preferred modality [15]. Plasma vancomycin levels and nephrotoxicity must be monitored very closely throughout treatment. The choice of the companion drug(s) depends on the susceptibility of the isolated strains, with rifampicin and minocycline being the drugs-of-choice in our experience [15,74].

Daptomycin is a valuable alternative option to treat methicillin-resistant PJIs and has less drug-related toxicity. Reported cure rates range between 54% and 87.5% [75,76,77]. Studies on daptomycin treatment of PJIs have usually been small, with heterogeneous populations (PJI types and surgical strategies), including susceptible and resistant strains and drug doses ranging from 4 to 8 mg/kg/day. As for vancomycin, daptomycin is combined with another effective antibiotic, most often rifampicin.

Linezolid is an option, when neither vancomycin nor daptomycin can be used. Despite its high bioavailability and in vitro efficacy against methicillin-resistant staphylococci [78], linezolid was the last choice to treat staphylococcal PJIs in a French national audit, probably because of its safety profile and the restricting nonuse beyond 4 weeks [71,78,79,80,81,82]. Study results showed increased tolerability of prolonged tedizolid administration [70,83], but data for broader use are lacking [84].

Dalbavancin has been prescribed to treat staphylococcal PJIs, but very small numbers of patients were enrolled in those studies and methicillin-susceptible strains were often included. Overall clinical cure rate was 73% [85].

Sheper et al. observed better outcomes of DAIR-treated CNS vs. *S. aureus* hip and knee PJIs in their systematic review and meta-analysis (73% vs. 62%, respectively) [63].

### 10.2. Streptococcal PJIs

Although staphylococci are the predominant pathogens isolated from PJIs, streptococci are the second most common microorganisms, found in 9–16% [6,7]. Streptococci can also form biofilm [86], but less is known about it than that of *S. aureus*. The majority of *Streptococcus* spp. that cause PJIs include group-B streptococci (*S. agalactiae*) and viridans-group streptococci [6,87]. Streptococci are more susceptible to antibiotics than *S. aureus*, but the question remains whether or not streptococcal PJI outcomes are better than staphylococcal infections. The results of a large study showed poor outcomes for DAIR-treated *Streptococcus* spp. PJIs [87]. Another multicenter cohort study on 70 streptococcal PJI patients identified *S. agalactiae* and DAIR as factors associated with failure [88].

β-Lactams are the first-choice treatment for streptococcal PJIs, except for penicillin-allergic patients. In their very large retrospective study on DAIR-treated streptococcal PJIs, Lora-Tamayo et al. observed that prolonged (≥21 days) β-lactam therapy was associated with therapeutic success. International guidelines recommend initial IV therapy with high-dose penicillin or ceftriaxone. Amoxicillin is also often given and is the first-choice treatment recommended by the French guidelines [89]. Initial IV administration and prolonged infusion of high doses, except for ceftriaxone, optimizes β-lactam time-dependent activity. For the oral switch amoxicillin is the first-line therapy (2–3 g, thrice daily).

Combination therapy with rifampicin remains controversial. Some authors use and recommend combination therapy with levofloxacin and rifampicin [90]. According to Lora-Tamayo et al. [87], rifampicin added no benefit in patients receiving a β-lactam, but was beneficial for those receiving a regimen without β-lactams. Mahieu et al. [88] did not observe better outcomes in patients receiving combination therapy with rifampicin or levofloxacin. International guidelines do not recommend combination therapy for streptococcal PJIs, regardless of the surgical strategy [5]. Combination therapy—especially including rifampicin—is associated with more AEs [43]. We obtained good results with β-lactam monotherapy, especially amoxicillin [91].

For penicillin-allergic patients, cefazolin, ceftriaxone, clindamycin, vancomycin or daptomycin are valid options for the initial IV therapy. Clindamycin rather than levofloxacin can be used for oral therapy, if the strain is susceptible. Linezolid should be reserved for when the other options are contraindicated.

For hematogenously spreading infections, identification and treatment of the infection source contributes to avoiding recurrent PJIs.

### 10.3. Enterococcus PJIs

*Enterococcus* spp. are less susceptible to penicillin than *Streptococcus* spp. Enterococcal PJIs have been associated with treatment failures and recurrent infections [92]. Polymicrobial infections are frequent (~50%) and associated with higher failure rates [92,93]. Therapeutic success rates vary widely and depend on the underlying comorbidities, surgical strategy applied and the mono- or polymicrobial nature of the infection [93,94,95].

Optimal treatment of these difficult-to-treat infections still has to be defined. International guidelines recommend high-dose IV penicillin or ampicillin. Combination therapy with gentamicin remains the prescriber’s choice [5]. In our experience and those of other teams who recently reported on these infections [96], high-dose IV amoxicillin is the first-choice treatment for amoxicillin-susceptible strains in nonallergic patients. Amoxicillin can be administered orally after 2–4 weeks of IV therapy. Ampicillin or penicillin is often combined with either gentamicin [94] or, more recently, rifampicin [92,96], but its benefit is still being discussed and AEs, such as nephrotoxicity after prolonged gentamicin administration or gastrointestinal intolerance, and drug interactions with rifampicin can occur. Amoxicillin–ceftriaxone, a valid alternative regimen for enterococcal endocarditis [97], was given in a small pilot study that included only 11 patients with various BJIs [98]. Nine of the 10 patients were cured, but only 3 patients were treated for PJIs, one received prolonged suppressive antibiotic therapy and follow-up duration was <2 years for five patients. Those limited data preclude drawing conclusions. Other combinations have been tried, such as daptomycin–fosfomycin or linezolid–rifampicin.

For resistant *Enterococcus* spp. or penicillin-allergic patients, IV high-dose vancomycin or daptomycin can be used. The oral regimen is limited to linezolid or the less toxic oxazolidinone, tedizolid.

Dalbavancin was administered every 2 weeks for 2–3 months to 16 patients with Gram-positive PJIs, including six enterococcal infections; none of those patients relapsed [85]. Larger studies are needed to confirm those findings.

### 10.4. Pseudomonas aeruginosa PJIs

Treatment of *P. aeruginosa* PJIs is considered difficult because of the bacterium’s natural resistance to many antimicrobials, increasing acquired resistance, the nosocomial environment and patients’ comorbidities [6,7,99]. The optimal antibiotic regimen and surgical strategy for *P. aeruginosa* PJIs are not well defined, and several questions remain. Agents with good activity against *P. aeruginosa* include ceftazidime, cefepime, piperacillin–tazobactam and carbapenem. Ciprofloxacin is the only available oral treatment for *P. aeruginosa*. Amikacin and tobramycin have intense bactericidal activity against *P. aeruginosa*.

Only a few retrospective studies focused on *P. aeruginosa* BJIs [99]. Recently, a French National Referral Center for BJIs reported on 90 implant-associated *P. aeruginosa* infections, among which 30 were PJIs [100]. Better outcomes were associated with at least 3 weeks of IV β-lactam administration and 3 months of oral ciprofloxacin. The overall cure rate was 77%. We reported the outcomes of our 43-patient *P. aeruginosa* PJI cohort [16]. Most were monomicrobial (*n* = 32), chronic, ciprofloxacin-susceptible infections (*n* = 35) and were treated with one-stage exchange (*n* = 27) and prolonged IV antibiotics. Overall, 82% percent of the patients and 93% of those managed with one-stage exchange had favorable outcomes.

International guidelines place cefepime or meropenem as the preferred agent, with ciprofloxacin and ceftazidime as alternatives. Adding aminoglycoside is treating physician’s choice; it can also be used locally in a spacer. Cefepime achieves high bone diffusion according to data from only one study [101].

Taking all these results together, prolonged (≥21 days) β-lactam combined with ciprofloxacin is certainly the treatment-of-choice for susceptible strains. In our recently reported experience, ceftazidime was the first-choice β-lactam, because of its excellent bactericidal activity against *P. aeruginosa* [102]. Cefepime, or piperacillin–tazobactam for polymicrobial infections, are other active agents. To optimize β-lactam’s time-dependent activity and reach therapeutic concentrations, continuous, high-dose infusion, TDM-adapted if possible, should be used. Aminoglycoside adjunction for the first few days can enhance bactericidal activity, followed by oral high-dose ciprofloxacin.

Although no standard treatment is recommended for ciprofloxacin-resistant *P. aeruginosa* PJIs, high-dose IV therapy is needed throughout the treatment duration. The potential benefit of combination therapy with other drugs (e.g., colistin, fosfomycin) has to be evaluated. Meropenem alone or in combination with other active drugs or ceftazidime–avibactam can be to treat multiresistant strain PJIs.

### 10.5. Enterobacteriaceae Infections

Enterobacteriaceae account for about 11–22% of PJIs [26,103]. They are also frequent components of polymicrobial infections. In two large retrospective multicenter cohort studies on Gram-negative PJIs, ~80% of the patients had favorable outcomes [26,103]. DAIR was the only surgical strategy in the Spanish study that included 242 PJIs, and DAIR (46%) or implant removal (54%) in the French investigation on 76 patients. In both studies, ~80% of the strains were ciprofloxacin-susceptible. The Spanish authors [26] underscored the crucial role of ciprofloxacin in treating these infections and the need for new therapeutic strategies to address fluoroquinolone-resistant organisms. The French authors [103] concluded that prolonged, IV β-lactam administration was an effective alternative to fluoroquinolones to treat these PJIs. In the latter study, 23.7% of the patients received nonfluoroquinolone regimens. High-dose IV β-lactams were administered with continuous cefepime and ceftazidime infusion. Those observations need to be confirmed by larger studies targeting a specific therapeutic strategy.

International guidelines for treating Enterobacteriaceae PJIs [5] recommend IV β-lactam selected according to in vitro susceptibility or oral ciprofloxacin 750 mg twice a day. Spanish and French guidelines [60,89] recommend starting high-dose IV cefotaxime or ceftriaxone and then prescribing oral ofloxacin or ciprofloxacin. Fluoroquinolones can be combined initially with the IV antibiotic. We adhere to those recommendations, starting with IV β-lactam, followed by oral fluoroquinolone. Ceftriaxone is the preferred drug because of its advantageous PK/PD characteristics. Effective trough target concentrations ~20 mg/L are easily achieved with one daily injection of 2 g that is also very convenient and well tolerated for parenteral outpatient antibiotic therapy [104]. Ciprofloxacin (500–750 mg, bid) or levofloxacin (750 mg once to 500 mg, bid) achieves higher AUC/MIC ratios and seems to be more effective than ofloxacin. When fluroquinolones cannot be used because of resistance or drug AEs, co–trimoxazole is an oral alternative. It is usually prescribed in combination, but robust data on its efficacy are limited [105]. We treat that setting with at least 4 weeks of ceftriaxone before switching to oral co–trimoxazole.

The prevalence of multidrug-resistant Enterobacteriaceae has been increasing for >20 years because of the broad variety of β-lactamases [106]. Many of these bacteria carry additional resistance genes, rendering them resistant to multiple drugs. To treat extended spectrum β-lactamase Enterobacteriaceae, Spanish guidelines recommend continuous IV carbapenem combined or not with colistin or fosfomycin [60] for the entire duration. Meropenem has the advantage of having lower MICs than imipenem and ertapenem, being administered by prolonged or continuous infusion, in contrast to imipenem, and being well tolerated even at high dose. In these complex treatment settings, tailoring therapeutic modalities to the antibiotic’s PK/PD characteristics, as described above, and dose needed to reach target concentrations based on MIC determination, are important tools to optimize treatment efficacy.

### 10.6. Cutibacterium PJIs

*Cutibacterium* spp. are highly susceptible to a wide range of antibiotics: ¦beta-lactams, clindamycin, rifampin, vancomycin and daptomycin [107,108]. However, increasing clindamycin resistance was reported [108]. *Cutibacterium* is naturally resistant to metronidazole. Currently available clinical data on these anaerobic opportunistic agents have not enabled definition of the optimal antibiotic therapy and no antibiotic-therapy consensus exists for *Cutibacterium* infections. Therapeutic regimens vary widely among countries and institutions. American guidelines [5] recommend penicillin or ceftriaxone as the first-choice treatment, with vancomycin or clindamycin as alternatives; French guidelines [109] recommend first-line amoxicillin, cefazolin or clindamycin. *Cutibacterium acnes* grows in biofilm and usually causes late, chronic PJIs, especially after shoulder arthroplasty [110,111]. The benefit of rifampicin combinations used by several teams, including ours in the past [112,113,114], has not been established by clinical study results and may be questioned when the implant is removed during exchange arthroplasty, which is the usual surgical strategy for these chronic infections. Our first-choice therapy is high-dose IV amoxicillin, clindamycin or cefazolin, followed by oral amoxicillin or clindamycin. For other teams [115], ceftriaxone, levofloxacin, linezolid and rifampicin are valid alternatives to use in combination. We used combination therapy with rifampicin until 2014 [114], and then changed to monotherapy [116] when no *Staphylococcus* was isolated, because of the lack of proven benefit of combination therapy and frequent AEs.

In patients with *Cutibacterium avidum* PJIs, we observed clindamycin-resistance in one-third and isolation of CNS or other anaerobes was also not rare [116]. For these reasons, we changed our procedures and recommend using vancomycin or daptomycin initially for *C. avidum* PJIs, while waiting for the final culture results of intraoperative samples.

### 10.7. Corynebacterium spp. Infections

*Corynebacterium* spp. PJIs are rare, around 1–2% [6,8], as are specific studies on them [117,118]. *Corynebacterium striatum* is the most frequently isolated species [117,119]. The choice of treatment for these difficult-to-treat infections relies on susceptibility testing, as antimicrobial efficacy is not predictable based on the species-level identification [120]. Amoxicillin is the drug-of-choice for susceptible strains for most teams. *Corynebacterium striatum* is almost always penicillin-resistant [117]. Noussair et al. recommend combining amoxicillin with rifampicin and reported curing 8 out of 12 monomicrobial *C. striatum* PJIs treated with various strategies [118]. Vancomycin is the treatment-of-choice before obtaining antibiotic susceptibility test results and for resistant strains or penicillin-allergic patients [118]. Daptomycin failure and development of drug resistance to it have been reported [119,120].

### 10.8. Culture-Negative PJIs

Culture-negative PJIs, ranging from 0% to >40%, represent an important and challenging issue for teams managing PJIs. The first step consists in establishing whether the cultures are truly negative and if the patient has a diagnosis other than a prosthesis infection. A false-negative may be attributable to prior antibiotic therapy, difficult-to-culture microorganism (anaerobic bacteria, *Mycoplasma hominis*, among others) or a microbe requiring special culture media, e.g., mycobacteria or fungi. A relatively recent literature review of the epidemiology, diagnosis and treatment of culture-negative PJIs emphasized that the definitive diagnostic method for identifying causative microorganism(s) has not yet been clearly described [121]. Based on their large multicenter observational study, Van Sloten et al. speculated whether all these infections should be treated with antibiotics [122]. Among their 1553 acute and 1556 chronic PJIs, only 4.7% of the chronic infections were culture-negative, a rate close to our experience of ~1% [8]. We agree with those authors that when adequate culture methods are used and antibiotics were withheld, even for several weeks in the case of chronic, indolent infections, the culture-negative infection rate was low. For patients whose diagnoses have not yet been established, implant sonication [123] and polymerase chain reaction gene-amplification techniques can help improve microbial identification. Next-generation-sequencing techniques, using DNA chips to search for numerous bacterial genes, are currently being evaluated [124]. In the end, the problem is essentially diagnostic rather than therapeutic.

Patients with culture-negative PJIs usually receive broad-spectrum antibiotics with activity against Gram-positive and Gram-negative pathogens. It was previously reported that they have outcomes similar to those of patients with positive cultures following standard treatment regimens [125]. As Van Sloten et al. [122] indicated, antibiotic treatment can probably be withheld from some patients, e.g., those with no histological signs of infection. For the others, identification of the causative pathogen should be attempted.

## 11. Antibiotic Prophylaxis

Antibiotic prophylaxis against PJI is still controversial [70,126]. In 2019, 30–39% of the teams included in a recent French national audit did not use it for PJI patients [70].

The authors of two studies concluded that antibiotic administration prior to intraoperative sampling did not impair culture sensitivity [127,128]. Wouthuyzen-Bakker et al. conducted a systematic review on the effect of preoperative antimicrobial prophylaxis on intraoperative sample-culture results of patients with suspected or confirmed PJIs [129]. They observed a 7% difference in culture positivity between those with antibiotic prophylaxis and without, and recommended maintaining preoperative antimicrobial prophylaxis, especially for patients undergoing revision arthroplasty with a low probability of infection.

The 2017 Spanish guidelines recommend appropriate antimicrobial prophylaxis throughout one-stage exchange arthroplasty [60]. We use antibiotic prophylaxis for revision arthroplasty in patients with confirmed PJIs, except those with a highly suggestive picture of chronic PJI, but without preoperative microbiological documentation based on one or more joint aspirations. We start antibiotic therapy during surgery within 1 h after incision, shortly after the first samples are taken. It remains debatable whether the use of high-dose “postoperative” antibiotics for a prolonged period of time, initiated rapidly during surgery, compensates, or not, for the single injection of preoperative antibiotic prophylaxis. The exact timing and duration of antibiotic prophylaxis after joint arthroplasty remain open; the overall grade of available evidence is low and randomized-control trials on that issue are still ongoing [130,131].

## 12. Preoperative Antibiotics

Preoperative antibiotics are usually not administered so as to assure the sampling of valid intraoperative samples for cultures. When the etiological diagnosis has been established by preoperative joint aspiration, especially for infections caused by *S. aureus* or Gram-negative bacillus, Spanish guidelines [60] recommend starting antibiotic therapy 3–5 days before one-stage exchange. We treat patients preoperatively when PJI has been confirmed and microbiologically documented by joint aspirate-culture results for patients meet the following criteria: fever or intense chills, bacteremia, presence of a large local inflammation or abscess and/or C-reactive protein ≥ 50 mg/L. Treatment choice is guided by results of the joint aspiration cultures. Its duration depends on the clinical picture and infection severity, the microorganism involved (at least 1 week for *S. aureus* PJI) and treatment strategy. In our cohort of 157 chronic PJIs treated with one-stage exchange arthroplasty, median preoperative therapy lasted 7 (IQR 4–11) days [56]. According to the recent French national audit, 12% of the teams started preoperative antibiotic therapy for late hematogenous *S. agalactiae* PJIs, based on joint aspiration-culture results [70].

## 13. Long-Term Suppressive Therapy

Prolonged suppressive antibiotic(s) with an oral regimen may be warranted for patients with progressive infection who are unsuitable for a curative procedure or refuse surgery. It is also a valid option for some patients with multiple recurrent PJIs [132]. This medical strategy is not further developed herein.

## 14. Perspectives

Future strategies for PJI management include phage therapy. It is still the exception, but intra-articular administration of selected bacteriophages has been combined with salvage DAIR surgery in patients with recurrent *S. aureus* and *P. aeruginosa* PJIs with a good clinical response [133,134]. However, prolonged suppressive antibiotic therapy has to be maintained.

Other local approaches attempt to improve the antimicrobial properties of inert Titanium orthopedic devices by using a multifunctional antimicrobial coating, also called biofunctionalized prosthesis. They combine passive antibacterial coating of the surface to prevent initial attachment of bacteria and an active antibacterial and immunomodulatory agent coating, which is released in the environment to improve the local immune and antimicrobial response [135]. Clinical data on these new approaches are warranted.

## 15. Conclusions

PJIs regroup various entities that differ by their postoperative or hematogenous contamination route, acute or chronic evolution, responsible microorganism(s) and their antibiotic susceptibilities, infection site and patient profiles. Choosing the optimal antibiotic regimen for a given patient must take all those parameters into consideration, together with the surgical strategy applied. Alternative regimens to the first-choice therapy are needed in case of adverse events or contraindication to certain drugs. The regimen for DAIR-treated MSSA PJIs is well codified. In contrast, many questions persist concerning the choice of antibiotics to treat PJIs caused by other pathogens. To address those unresolved issues, future studies should be designed to target well-defined settings, i.e., type of infection, microbe(s) and surgical strategy.

## Figures and Tables

**Figure 1 antibiotics-11-00486-f001:**
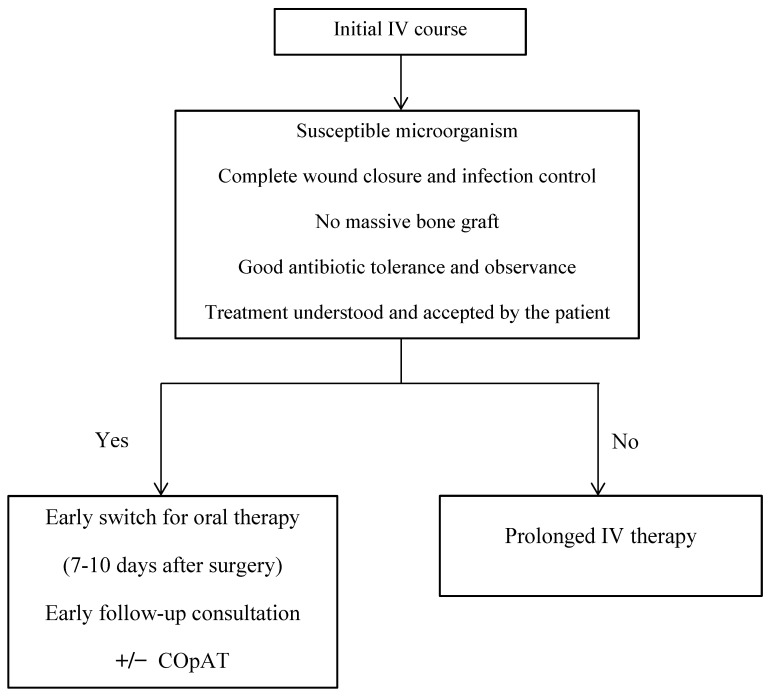
Initial intravenous therapy and indications to oral switch. IV: intravenous; COpAT: complex outpatient antibiotic therapy.

**Figure 2 antibiotics-11-00486-f002:**
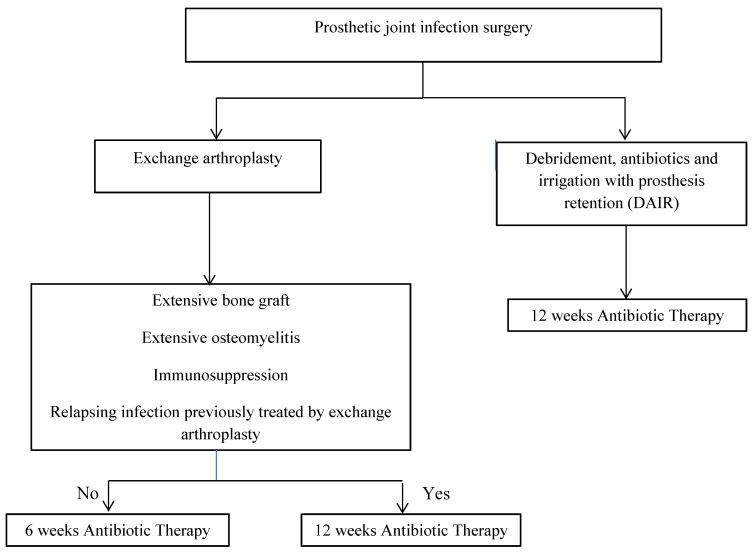
Duration of antibiotic therapy.

**Table 1 antibiotics-11-00486-t001:** Indications, advantages and risks of IV versus oral antibiotic therapy.

	Intravenous Antibiotic Therapy	Oral Antibiotic Therapy
**Indications**	Initial PJI treatment Use of drugs not orally available Treatment of resistant or difficult to treat microorganism	Oral switch after initial IV therapy Suppressive antibiotic therapy
**Advantages**	Allows use of high drug dosages PK/PD optimization (extended or continuous infusion) 100% bioavailability (avoids nonobservance, malabsorption and hepatic first-pass effect) Better gastro-intestinal tolerance Better monitoring during hospitalization or outpatient parenteral antibiotic therapy	No venous access No risk of catheter-related complication Better rehabilitation and mobility
**Risks** **Disadvantages**	Venous catheter-related complications (infection, thrombosis,…) Reduced mobility Longer hospital stay	Lower bioavailability due to drug-specific absorption and hepatic first effect Frequent gastro-intestinal intolerance Possible non observance More parameters that can vary antibiotic serum concentrations

**Table 2 antibiotics-11-00486-t002:** Pathogen-specific first-choice antibiotic therapies.

Microorganism	Initial IV therapy	Oral switch
Methicillin-susceptible *Staphylococcus*	Cefazolin or oxacillin + rifampicin	Levofloxacin + rifampicin
Methicillin-resistant *Staphylococcus*	Vancomycin or daptomycin + another drug depending on the strains’ susceptibility (rifampicin or minocycline or linezolide)	No oral switch
*C. acnes*	Amoxicillin or clindamycin (check clindamycin susceptibility)	Amoxicillin or clindamycin
*Streptococcus*	Amoxicillin	Amoxicillin
*Enterococcus faecalis*	Amoxicillin ± initial gentamicin	Amoxicillin
*Corynebacterium*	Susceptible strain: amoxicillin Resistant strain: vancomycin ± rifampicin	Susceptible strain: amoxicillin
*Enterobacteriaceae*	Ceftriaxone or cefotaxime	Ciprofloxacin or levofloxacin
*Pseudomonas aeruginosa* (ciprofloxacin susceptible strain)	Ceftazidime or cefepime + initial amikacine followed by ceftazidime or cefepime + oral ciprofloxacin At least 3 weeks	Ciprofloxacin

## Data Availability

Not applicable.

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
