# Peer review of "Antibiotic Therapy for Prosthetic Joint Infections: An Overview"

_antibiotics, 2022, doi:10.3390/antibiotics11040486_

Round 1
Reviewer 1 Report
I am delighted to review this review article on Prosthetic Joint Infections therapy and its management. The manuscript follows the scope of the journal Antibiotics. A detailed literature survey has been done with an excellent presentation. A flow chart of the topic going to be discussed in this review at the beginning of the discussion will have a good overview for the readers.
I would recommend the article could be published in Antibiotics with minor corrections. And the authors need to address the below-mentioned queries.
(a) The author could mention the global number for following “Nearly 200,000 total hip replacements are performed annually in France.”
(b) Introduction is a little bit lengthy; the author could try to reduce it.
(c) For “Initial IV Antibiotics”: The author could provide 2-3 examples showing the advantage of Intravenous administration over oral delivery in the table.
(d) For “Extended Infusion”: The author could provide bone/plasma ratio of other drugs along with as provided for cefazolin if data are available.
(e) If the author could provide a graph for comparison of therapies for the section “4. Oral vs. IV Therapy” that would make this section clearer.
(f) A graphical presentation of results for the “Antibiotic-Treatment Duration” could increase the visuality of the discussion. The author could provide data in tabular or graphical format for all sections whenever applicable.
(g) For “Pathogen-Specific Antibiotic Therapies”: the author could provide a flow chart.
(h) Change the color of the text of the references (2, 3, 59, and 107) from blue to black. Change red color letter “o” from reference 130 to black. Change green color letter “e” from reference 104 to black.
(i) The author could include the following relevant references:
(a) Sousa R, Pereira A, Massada M, da Silva MV, Lemos R, Costa e Castro J. Empirical antibiotic therapy in prosthetic joint infections. Acta Orthop Belg. 2010 Apr;76(2):254-9. PMID: 20503953.
(b) Sousa R, Abreu MA. Treatment of Prosthetic Joint Infection with Debridement, Antibiotics and Irrigation with Implant Retention - a Narrative Review. J Bone Jt Infect. 2018;3(3):108-117. Published 2018 Jun 8. doi:10.7150/jbji.24285
(c) https://doi.org/10.1111/imj.15677
Author Response
Reviewer 1
- The author could mention the global number for following “Nearly 200,000 total hip replacements are performed annually in France.”
Answer of the author : Sorry but there is no arthroplasty register in France and the reference studied and mention only hip arthroplasties. We didn’t change that point for those reasons.
- Introduction is a little bit lengthy; the author could try to reduce it.
Answer of the author : You are right. We shortened the introduction.
- For “Initial IV Antibiotics”: The author could provide 2-3 examples showing the advantage of Intravenous administration over oral delivery in the table.
Answer of the author : We added a table listing the indications, advantages and risks of IV and oral treatment. See table 1.
- For “Extended Infusion”: The author could provide bone/plasma ratio of other drugs along with as provided for cefazolin if data are available.
Answer of the author : Sorry but we have no data available on other drugs.
- If the author could provide a graph for comparison of therapies for the section “4. Oral vs. IV Therapy” that would make this section clearer.
Answer of the author : We added a table listing the indications, advantages and risks of IV and oral treatment and a figure with the indications of the oral switch or prolonged IV antibiotic therapy. See table 1 and figure 1.
- A graphical presentation of results for the “Antibiotic-Treatment Duration” could increase the visuality of the discussion. The author could provide data in tabular or graphical format for all sections whenever applicable.
Answer of the author : We added a figure listing the different antibiotic duration according to the surgical strategy. See Figure 2.
- For “Pathogen-Specific Antibiotic Therapies”: the author could provide a flow chart.
Answer of the author : We added a table listing the first choice pathogen specific antibiotic therapies. See Table 2.
- Change the color of the text of the references (2, 3, 59, and 107) from blue to black. Change red color letter “o” from reference 130 to black. Change green color letter “e” from reference 104 to black.
Answer of the author : Done.
- The author could include the following relevant references:
Answer of the author : Done.

Reviewer 2 Report
The authors carried out a detailed review on antibiotic therapies for PJI management. They mainly described intravenous and oral antibiotic therapies. They point out the relevance of defining the treatment duration and considering pharmacokinetic and pharmacodynamic features of drugs, among other factors. The authors also remark the difficulties associated with biofilm formation and suggest the determination of plasma drug concentration as a helpful tool to optimize antibiotic therapies. Finally, they discussed the specific antibiotic therapies for specific pathogens.
I consider the paper can be accepted in present form. Nevertheless, I make the following suggestions/comments:
- Unification of section 2 (Initial IV Antibiotics) and section 3 (Extended Infusion) for better following the discussion.
- Introduce a section to describe future trends on the topic. I also consider it could be of interest to briefly describe local treatments that employs biofunctionalized prosthesis loaded with antibiotics.
Author Response
Reviewer 2
- Unification of section 2 (Initial IV Antibiotics) and section 3 (Extended Infusion) for better following the discussion
Answer of the author : Done.
- Introduce a section to describe future trends on the topic. I also consider it could be of interest to briefly describe local treatments that employs biofunctionalized prosthesis loaded with antibiotics
Answer of the author : We added a short last paragraph on future local treatments including phage therapy and biofunctionalized prosthesis and added 3 references [133-135].
- Perspectives
Future strategies for PJI management include phage therapy. It is still the exception, but intra articular administration of selected bacteriophages has been combined with salvage DAIR surgery in patients with recurrent S. aureus and P. aeruginosa PJIs with a good clinical response [133, 134]. However, prolonged suppressive antibiotic therapy has to be maintained.
Other local approaches attempt to improve the antimicrobial properties of inert Titanium orthopedic devices by using multifunctional antimicrobial coating, also called biofunctionalized prosthesis. They combine passive antibacterial coating of the surface to prevent initial attachment of bacteria and active antibacterial and immunomodulatory agent coating which will be released in the environment to improve the local immune and antimicrobial response [135]. Clinical data on these new approaches are warranted.